# Revisiting the cognitive and behavioral aspects of loneliness: Insights from different measurement approaches

Andrej Skoko[1]*, Noëmi Seewer[1], Marcus Mund[2], Tobias Krieger[1]

1 Department of Clinical Psychology and Psychotherapy, University of Bern, Bern, Switzerland,
2 Department of Personality Psychology and Psychological Assessment, University of Klagenfurt, Klagenfurt, Austria

* andrej.skoko@unibe.ch

**Data availability statement:** All relevant data are available on the Open Science Framework (https://osf.io/ftr6b/).

**Funding:** TK grant number: 100019_192416 Swiss National Science Foundation www.snf.ch. The funding body played no role in the study design, data collection, or manuscript writing.

## Abstract

Loneliness is increasingly recognized as a critical public health issue that profoundly affects psychological well-being and social functioning. This study evaluates cognitive and behavioral differences associated with different facets of loneliness. We classified 790 German-speaking adults ($M_{Age}$ = 31.86 (12.48), 81% female) as lonely or not lonely based on three dimensions - loneliness frequency, distress, and chronicity - and tested for group differences regarding cognitive and behavioral aspects, as proposed by the cognitive model of loneliness, while controlling for depressive and social anxiety symptoms. The results indicate fair to substantial agreement between the three classification methods. Further, we found significant group differences regarding all components, such as interpretation bias, social avoidance, and self-esteem, with each loneliness classification method. Our findings highlight the multifaceted nature of loneliness and underscore the importance of applying diverse methods to fully capture its complexity. This study contributes to a more nuanced understanding of loneliness and its implications, suggesting that interventions should consider the specific dimensions of loneliness to effectively address its cognitive and behavioral ramifications.

## 1. Introduction

Loneliness is defined as a subjectively distressing emotional state characterized by a discrepancy between the desired and perceived quality or quantity of social relationships [1]. While nearly everyone experiences loneliness at some point in their life, for some, it can become an enduring condition with potentially negative implications for mental and physical health and even increased mortality [see for reviews 2–4], elevating loneliness as a global health priority [5]. While the COVID-19 pandemic led to increased loneliness, the effect size of this change was small [6]. Still, pre-pandemic data from 113 countries showed loneliness prevalence ranging from 5.3% to 12.7% across age groups [7], emphasizing the need for continued research and intervention.

**Competing interests:** The authors have declared that no competing interests exist.

## 1.1. Adaptive and maladaptive loneliness

Although increased levels of loneliness can have negative health effects, short-term loneliness can prompt individuals to seek out social connections [8]. From an evolutionary standpoint, loneliness is considered to function as a cue indicating endangered social relationships, which should encourage behaviors aimed at reconnecting with existing relationships or establishing new ones [9,10]. Therefore, loneliness can act as a healthy and adaptive response, signaling the need for changes in one's social life.

In their recent review, Maes and Vanhalst [11] show in their recent review that previous theoretical frameworks and research argue that loneliness in its prolonged/chronic and consequently maladaptive state can be particularly concerning due to its association with cognitive biases and behavioral tendencies, which are described in the cognitive model of loneliness by Cacioppo and Hawkley [12]. This model suggests that loneliness triggers a cascade of cognitive processes that heighten awareness of social disconnection. These processes include heightened sensitivity to subjective social threats, negative attributions, and biased social information processing, which can lead to maladaptive behaviors such as social withdrawal and increased vigilance toward potential social threats and thereafter maintain or increase feelings of loneliness [10,12]. While Qualter et al. [10] discuss how individuals in this state of hypervigilance can escape these maladaptive tendencies and show more adaptive behaviors with interventions, the question remains unclear what cognitive and behavioral patterns are associated with adaptive loneliness. Nonetheless, this distinction between adaptive and maladaptive loneliness highlights the importance of considering not just the presence of loneliness but its persistence, intensity, and cognitive and behavioral consequences. Understanding what maladaptive and adaptive experiences of loneliness encompass remains a critical area for further research.

## 1.2. Loneliness and distorted social information processing

In trying to examine empirical evidence for the cognitive model of chronic loneliness [12], Spithoven et al. [13] used the social information processing (SIP) model [14] in their comprehensive review on cognitive biases in lonely individuals, that affect various stages of social information processing, from attention to interpretation and response selection in social situations. They highlighted the tendency of lonely individuals to interpret social information in a negative light, anticipate rejection, and have negative self and others' evaluations, the increased pursuit of avoidance goals, heightened social avoidance/withdrawal behaviors, and fewer social skills, among others [for the detailed review, see 13]. More recent findings support the reported tendencies regarding negative interpretation bias [e.g., 15–17], higher rejection sensitivity [e.g., 18,19], negative self-evaluation and low self-esteem [e.g., 20,21], increased social avoidance behavior [e.g., 22], and avoidance motivation [e.g., 23]. All in all, these findings support the hypothesis that lonely people show distorted social information processing in different areas.

### 1.3. Bias information processing related to symptoms of depression and social anxiety

Although loneliness is distinct from depression and social anxiety [e.g., 24], it is strongly associated with both [e.g., 25]. Since these conditions also involve biased information processing [for depression see 26, for social anxiety see 27], it is crucial to account for depressive and social anxiety symptoms when examining cognitive and behavioral differences in lonely individuals. In this study, we wanted to test whether differences regarding social information processing [cf., 13] are maintained when controlling for psychopathological symptoms such as depressive symptoms and symptoms of social anxiety.

### 1.4. Assessing loneliness

Due to the complexity and subjective nature of loneliness, it is challenging to measure it accurately. Most studies thus far used scales such as UCLA Loneliness Scale [UCLA-LS; 28] or, the Rasch-Type Loneliness Scale [RTLS; 29] (also known as the De-Jong Gierveld Scale) to assess loneliness. While such scales seem to be a robust tool for measuring general loneliness, their predominant use has limitations. According to Maes et al. [30], many items in loneliness scales, including the UCLA-LS and the RTLS, may not directly measure loneliness as defined by Peplau and Perlman [1] (i.e., the discrepancy between desired and actual social relationships), but are rather related or predictive of loneliness to some extent. Moreover, most loneliness measures do not include any timeframe with which reported feelings of loneliness can be referenced [31]. These points can potentially compromise the validity of the findings.

Qualter et al. [31] state that current research often focuses on the frequency of loneliness, for instance, when using most English versions of the UCLA-LS. Other versions of the UCLA-LS [e.g., 32] or the RTLS [29] (also known as the De-Jong Gierveld Scale) use items with categories that reflect agreement [30], which might be interpreted as the intensity. However, Qualter et al. [31] argue that current measures might primarily address the persistence of loneliness-related emotions and behavior and that exploring its severity may be more effectively conceptualized through intensity (e.g., by explicitly asking about the distress connected to loneliness) or duration (e.g., by asking how long feelings of loneliness lasted), or a combination of these measures. They further point toward that Weiss [33] argued that both the frequency and intensity of loneliness should be examined. Despite this longstanding recognition of the importance of frequency, intensity, and duration, there is little exploration into which measures best indicate the severity of loneliness, and most scales were developed without this consideration [31].

Direct measurement of loneliness involves asking respondents explicitly if they feel lonely, using one-item questions such as "Do you feel lonely?" which are mostly used in epidemiological studies. Indirect measurements such as the UCLA-LS are multiple-item scales that do not explicitly use the word loneliness. Shiovitz-Ezra and Ayalon [34] found significant discrepancies between direct and indirect measures of loneliness, with more than half of respondents who reported loneliness on the direct measure but were classified as not lonely on the indirect measure. Similarly, Nicolaisen and Thorsen [35] argue that different prevalence rates might emerge due to the heterogeneity of loneliness measures (direct vs. indirect).

However, the direct single-item measures of loneliness might fail to fully capture the subjective distress of loneliness. To address this, Reinwarth et al. [36] formulated a single item, which tried to include feelings of distress regarding loneliness to provide a better screening tool for large-scale population study. The single-item was formulated as "I am frequently alone/have few contacts" and could be rated as 0 "no, does not apply", 1 "yes, it applies, but I do not suffer from it", 2 "yes, it applies, and I suffer slightly", 3 "yes, it applies, and I suffer moderately", or 4 "yes, it applies, and I suffer strongly". They then summarized the responses of the participants [similar to 37] and recoded loneliness by combining 0 and 1 "no loneliness or distress", 2 "slight loneliness", 3 "moderate loneliness", and 4 "severe loneliness." The comparison with the three-item version of the UCLA-LS showed similar prevalence rates. They further found a moderately positive correlation ($\rho$ =.57, $p$ <.001), which stands in line with previous comparisons between direct single-items and the three-item version of the UCLA-LS [38]. Despite this comparability, these results further point toward a remaining divergence regarding

measurements of loneliness. Here, it can be argued that the single-item might focus on the distress (or intensity) and the UCLA-LS on the frequency of loneliness.

Mund et al. [38] tested different multi- and single-item (direct and indirect) measures for loneliness and demonstrated that single-item measures of loneliness mostly have high correlations with each other and with multi-item scales, and their nomological nets align with established measures like RTLS and UCLA-LS. However, some differences emerged from the nomological nets: the UCLA-LS tended to have higher positive correlations with constructs such as neuroticism and depressiveness and negative correlations with, for instance, extraversion life satisfaction compared to the direct single-item [38]. While this might suggest that indirect measures might capture broader aspects of loneliness, they might have lower specificity than direct approaches. Nevertheless, they also showed that single-item measures are reliable, validating their use as robust tools in loneliness research [38].

In sum, these similarities and differences between measuring methods highlight that direct and indirect approaches may capture overlapping facets of loneliness but might also assess diverging aspects, which can lead to variability in the reported prevalence and associated characteristics of lonely individuals. This indicates that it might be necessary to account for different facets for a more holistic view of loneliness.

### 1.5. Defining maladaptive loneliness

When addressing a maladaptive form of loneliness [11], there is considerable heterogeneity in the literature. Often, terms such as "prolonged" or "chronic" loneliness are used to distinguish a maladaptive from an adaptive form [e.g., 39]. However, some scholars define maladaptive forms of loneliness based on frequency [e.g., 40], others based on distress associated with loneliness [e.g., 37], and others based on chronicity [e.g., 41]. To counter this heterogeneity, recent research has increasingly emphasized the importance of assessing loneliness as a multidimensional construct, incorporating different aspects [e.g., 31,42]. These efforts align with the broader need to refine loneliness measurement and account for its diverse manifestations.

In the context of specifically investigating chronic loneliness, the definition of Young [43] has been used as a reference, which characterizes chronic loneliness as loneliness persisting for two years and beyond. Older cross-sectional studies relied on the self-reported duration of loneliness to operationalize chronic loneliness [44–46]. As previously mentioned, Qualter et al. [31] pointed out that most loneliness scales, which are frequently used in recent studies, typically ask participants about the frequency of their loneliness, using Likert-type scales from 'never' to 'always.' These scales are then also applied in longitudinal studies to classify participants into 'not lonely,' 'temporary lonely,' and 'chronically lonely' groups, where the occurrence of a certain level of loneliness at all time points determines the affiliation to one of the categories [e.g., 47,48–52]. These study designs seem to integrate both duration and frequency aspects. However, one potential issue emerging from this is that it is unclear how well chronic loneliness is encompassed in these studies since changes in loneliness between the time points were not captured.

Maes and Vanhalst [11] suggest that future research should disentangle the duration, frequency, and intensity of loneliness and investigate how these aspects, both individually and in combination, relate to, for instance, health outcomes. Qualter et al. [31] have already implemented this multi-dimension approach by using an adapted four-item version of the UCLA-LS, where they assessed the frequency, intensity, and duration. For instance, regarding the duration, the item "Do you feel a lack of companionship?" was followed by the question "How long does that feeling last when it occurs?" and answered with these response options "1 = hours, 2 = days, 3 = weeks, 4 = months, 5 = longer". They then employed Latent Class Profile Analyses to identify distinct groups of individuals based on their loneliness experiences. The analysis revealed four groups, each characterized by different levels of loneliness across the three measures. One key finding was that the duration of loneliness, particularly experiences lasting months or years, was critical in distinguishing between these groups [31]. This approach highlights the importance of considering multiple dimensions of loneliness to understand its impact fully.

### 1.6. Current study

Our study aims to investigate whether the cognitive and behavioral differences between lonely and non-lonely individuals persist when comparing groups of lonely and non-lonely people based on different aspects of loneliness. Specifically, we first applied three different methods of classifying high vs. low lonely individuals: an indirect measurement of loneliness, a direct measurement of loneliness and the associated distress, and chronicity of loneliness. Then, we test the degree of agreement between these three methods, where we expect to find at least a moderate agreement. Finally, we examine group differences between high vs. low lonely participants for each of the different classification methods regarding different cognitive or behavioral aspects of the cognitive model of chronic loneliness, i.e., regarding interpretation bias, rejection sensitivity, social avoidance behavior, distress disclosure, self-esteem, and avoidance goal intensity, while controlling for depressive and social anxiety symptoms. We expect to find significant differences between lonely vs. non-lonely groups regarding all variables. However, this study is mainly exploratory in nature for the two latter classifications since they are rather new.

## 2. Methods

A total of 1,389 individuals initially accessed the online survey. Of these, 553 participants (39.8%) did not finish the survey and were excluded from the final sample. To ensure data integrity, bogus items were implemented in the survey, where another 23 participants were excluded. Two participants who reported being under the age of 18 were also excluded. Finally, we used complete-case analysis, leading to the exclusion of an additional 21 participants. The final sample consisted of 790 German-speaking adult participants.

Participants were recruited from the general population between February 2021 and March 2022 through online platforms such as SurveyCircle, social media, and internet forums (e.g., www.psychic.de). The survey was titled "Survey Study on Loneliness" and was administered online using Qualtrics (Qualtrics XM). Written informed consent was obtained from all participants prior to participation. The study was approved by the ethics committee of the Faculty of Human Sciences at the University of Bern (2020-08-00005).

To participate, individuals had to be 18 years or older and proficient in German. The majority of the final sample were women (81%), with a mean age of 31.86 years (SD = 12.48, range = 18–90). 56.1% were employed, and 47.3% held a university or university of applied sciences degree. In terms of relationship status, 41.7% were single, 52.3% were in a relationship or married, and 3.4% were divorced or widowed.

### 2.1. Measures

**2.1.1. Loneliness.** We assessed high vs. low lonely individuals in three different ways. First, we used the most commonly used way to classify lonely vs. non-lonely individuals by a cut-off applied to an indirect measure of loneliness, i.e., a short version UCLA Loneliness scale, which assesses foremost the frequency of loneliness [see 31]. Second, we created groups of lonely vs. non-lonely individuals based on the distress that is linked to loneliness. Third, we created groups applying the 2-year criterion proposed by Young [43] to assess the chronicity of loneliness.

*Loneliness frequency* was measured using the German 9-item version [53] of the UCLA loneliness scale [UCLA-LS; [28,54]. Sample items are: "How often do you feel that you lack companionship?", "How often do you feel that you have a lot in common with the people around you?". The items were rated on a Likert scale from 1 (never) to 4 (always). Higher scores indicate increased loneliness. The internal consistency in our sample was high (see Table 1). They were asked how often these statements applied to them in the last four weeks. We dichotomized the scale (1 = lonely vs. 0 = not lonely) with a cut-off point of ≥ 27 for the analyses, similar to Shiovitz-Ezra and Ayalon [34].

*Loneliness distress* was operationalized by using two single-items. Loneliness was assessed with a single direct question ("Do you feel lonely?"; rated on a 4-point scale with the response options 0 = "no, never"; 1 = "yes, sometimes"; 2= "yes, quite often"; 3 = "yes, very often"). Further, we assessed the feeling of distress caused by loneliness ("To what

**Table 1. Descriptive statistics and zero-order correlations, including Cronbach's alpha.**

| Variables | M | SD | Mdn | Range | 1 | 2 | 3 | 4 | 5 | 6 | 7 | 8 | 9 | 10 | 11 |
|---|---|---|---|---|---|---|---|---|---|---|---|---|---|---|---|
| Loneliness Frequency | 0.16 | 0.37 | 0.00 | 0.00-1.00 | (.89) | | | | | | | | | | |
| Loneliness Distress | 0.23 | 0.46 | 0.00 | 0.00-1.00 | .43*** | – | | | | | | | | | |
| Chronic Loneliness | 0.30 | 0.50 | 0.00 | 0.00-1.00 | .31*** | .68*** | – | | | | | | | | |
| Interpretation Bias | 1.84 | 0.61 | 1.75 | 1.00-4.00 | .32*** | .31*** | .28*** | (.63) | | | | | | | |
| Rejection Sensitivity | 9.64 | 4.69 | 9.00 | 1.38-36.00 | .41*** | .43*** | .34*** | .55*** | (.83) | | | | | | |
| Social Avoidance Behavior | 19.57 | 6.73 | 19.00 | 8.00-40.00 | .30*** | .29*** | .28*** | .47*** | .50*** | (.87) | | | | | |
| Distress Disclosure | 38.29 | 10.36 | 38.00 | 4.00-60.00 | -.24*** | -.17** | -.14** | -.21*** | -.28*** | -.37*** | (.93) | | | | |
| Self-Esteem | 18.39 | 7.09 | 19.00 | 0.00-30.00 | -.37*** | -.47*** | -.37*** | -.48*** | -.55*** | -.45*** | .30*** | (.92) | | | |
| Avoidance Goal Intensity | 3.66 | 0.55 | 3.69 | 1.46-5.00 | .14*** | .30*** | .20*** | .14*** | .20*** | .08* | -.03 | -.30*** | (.83) | | |
| Depressive Symptoms | 10.22 | 6.21 | 9.00 | 0.00-27.00 | .35*** | .52*** | .40*** | .34*** | .46*** | .39*** | -.21*** | -.69*** | .28*** | (.88) | |
| Social Anxiety Symptoms | 6.36 | 5.10 | 5.00 | 0.00-24.00 | .35*** | .37*** | .31*** | .50*** | .53*** | .63*** | -.28*** | -.53*** | .21*** | .52*** | (.85) |

*Notes.* Cronbach's alphas are provided in parentheses on the diagonal.

*p <.05,

**p < 0.01,

***p <.001. N = 790.

extent do you feel distressed by the stated feelings of loneliness?"; 0 = "not at all"; 1 = "a little"; 2 = "quite"; 3 = "strongly"; 4 = "very strongly"). Then, we created a dichotomous variable to combine the two variables similar to Reinwarth et al. [36]: a combination of values from 2 to 3 on the loneliness item and values from 2 to 4 on the distress item were coded as 1 (lonely) and the rest as 0 (not lonely).

*Loneliness chronicity* was operationalized by again using the above-mentioned direct single-item question. For participants who reported feeling lonely at least quite often, we asked for how many months this was already the case. We then created a dichotomous variable (1 = lonely and 0 = non-lonely) with a cut-off value of 24 months in accordance with the definition by Young [43], which has been used in previous cross-sectional [44–46] and longitudinal studies [e.g., [47,48–52].

**2.1.2. Cognitive and behavioral variables related to loneliness.** As described above, Spithoven et al. [13] demonstrated in their review that lonely individuals seem to exhibit cognitive and behavioral differences in regard to social information processing compared to non-lonely individuals. To assess several of those components, we used the following questionnaires:

*Interpretation bias* in socially ambiguous situations was assessed with the respective subscale of the Interpretation and Judgmental Questionnaire [IJQ; [55,56]. The scale was used in previous studies assessing interpretation bias [56–58]. The scale consists of social events with positive, ambivalent, mildly negative, or profoundly negative valence. Five brief vignettes were presented for each valence. Four interpretations for every event were used as the response format, ranging from positive, ambiguous, and mildly negative to profoundly negative, which the participants had to rate for plausibility ("Which of the four answers seems most plausible/appropriate to you?") by ranking them from one (most plausible) to four (least plausible). We used the subscale of ambivalent situations for the analyses. First, the mean rank of the profoundly negative interpretation was calculated over situations. The score is the mean rank given to the profoundly negative interpretation of the scenarios and ranges between 1 and 4. We reverse-coded the ranks, meaning a higher score indicates more negatively biased processing. The internal consistency in our sample was moderate for the socially ambiguous situations (see Table 1).

*Social avoidance behavior* was measured with a subscale from the Cognitive-Behavioral Avoidance Scale [CBAS; [59,60]. For this study, the 8-item behavioral social subscale was used (e.g., "I tend to make up excuses to get out of social activities," "I avoid attending social activities"). The rating consisted of a five-point Likert scale (from 1 = "not at all

true for me" to 5 = "completely true for me"), with higher scores indicating increased social avoidance behavior. The internal consistency in our sample was high (see Table 1).

*Rejection sensitivity* was assessed using the adapted adult version [A-RSQ; 61] of the Rejection Sensitivity Questionnaire [62]. In the A-RSQ, 9 hypothetical interpersonal situations are presented, and respondents indicate how they would feel or think in the stated situations. Participants indicated on a 6-point scale how concerned they would be in that situation (from 1 = "very unconcerned" to 6 = "very concerned") and how likely they would expect to be accepted (from 1 = "very unlikely" to 6 = "very likely"). Those two responses were then multiplied for each scenario, and afterward, a mean score was calculated with higher values indicating higher rejection sensitivity. The internal consistency in our sample was high (see Table 1).

*Comfort with self-disclosure* was assessed using the Distress Disclosure Index [DDI; 63]. It is a 12-item scale designed to measure the degree to which a person is comfortable talking with others about personally distressing information (e.g., "I am willing to tell others my distressing thoughts"). Items are rated on a 5-point Likert-type scale (1 = "strongly disagree" to 5 = "strongly agree"). The sum score was used for the analyses with higher values indicating more comfort with self-disclosure. The internal consistency in our sample was high (see Table 1).

*Self-esteem* will be assessed using the 10-item revised German version [54] of the Rosenberg Self-Esteem Scale [64] of the Rosenberg Self-Esteem Scale [RSES; 65]. This scale measures the positive and negative aspects of self-esteem. Items are rated on a 4-point Likert scale (0 = "strongly agree" to 3 = "strongly disagree"). The sum score was used for the analyses, and higher values indicated higher self-esteem. The internal consistency in our sample was high (see Table 1).

*Avoidance Goal Intensity* was assessed with the Inventory of Approach and Avoidance Goals [IAAM; German: Fragebogen zur Analyse Motivationaler Schemata [FAMOS]; 66]. The original IAAM consists of 94 items; 57 assess the intensity of approach goals, and 37 the intensity of avoidance goals. In the current sample, we only assessed the avoidance goals of the subscales "Aloneness/Separation," which has five items (e.g., "not receiving enough love and attention"), "Deprecation/Derogation," consisting of five items (e.g., "not being respected"), and "Vulnerability," which comprises of three items (e.g., "to show your own weaknesses"). Items are rated on a 5-point Likert scale (1 = "not at all terrible" to 5 = "extremely terrible"). The mean over the 13 items was used for the analyses, with higher values indicating higher avoidance goal intensity. The internal consistency in our sample was high (see Table 1).

**2.1.3. Psychopathological symptoms.** Based on the evidence that depression [26] and social anxiety [27] are related to biased information processing and the potential that differences in information processing between lonely and non-lonely individuals might emerge from differences with regard to depressive and social anxiety symptoms, we used the following questionnaires to assess depressive and social anxiety symptoms as covariates:

*Depressive symptoms* were assessed with the 9-item depression module of the Patient Health Questionnaire [PHQ-9; [67,68]. All nine items correspond to the nine DSM-IV criteria for depression. The items are rated on a 4-point Likert scale (from 0 = "not at all" to 3 = "nearly every day"). The sum score ranging from 0 to 27 was used for the analysis, with higher scores reflecting higher levels of depressive symptoms.

*Symptoms of social interaction anxiety* are measured with the German translations of the short-form of the Social Interaction Anxiety Subscale [SIAS-6; 69]. The six items are rated on a 5-point Likert scale (from 0 = "not at all" to 4 = "extremely"). The sum score ranging from 0 to 24 was used for the analysis, with higher scores reflecting higher levels of social interaction anxiety symptoms.

**2.1.4. Questionnaire order and administration.** All questionnaires were presented in a fixed order to maintain consistency. The order was as follows: (1) demographic information, (2) loneliness measures, (3) cognitive and behavioral variables, and (4) psychopathological.

## 2.2. Statistical analyses

All analyses were conducted in R [70]. First, we calculated the bivariate correlations of all measures. Second, we divided the sample in three ways: lonely vs. not lonely using the ≥ 27 cut-off of the UCLA-LS, distressed vs. not distressed due to

loneliness using the two single items, and chronically lonely vs. not chronically lonely using the self-reported duration of loneliness with the 24-month cut-off. Third, we calculated Pearson's $X^2$-tests for gender and Welch's two-sample t-tests for age, depressive, and social anxiety symptoms to see if there are significant group differences. Fourth, we tested the degree of agreement between the three classification methods using $\chi^2$ analysis, which is appropriate for assessing categorical variable associations. Since loneliness classifications in this study are dichotomous (lonely vs. non-lonely), $\chi^2$ allows us to determine whether the distributions of participants across methods significantly differ beyond random chance. To further quantify agreement, we computed percentages, Cohen's $\kappa$ [71], and $r_\varphi$. Cohen's $\kappa$ values are interpreted as follows: values ≤ 0 indicate no agreement, 0.01–0.20 indicate slight agreement, 0.21–0.40 indicate fair agreement, 0.41–0.60 indicate moderate agreement, 0.61–0.80 indicate substantial agreement and 0.81–1.00 indicate almost perfect agreement [72]. $r_\varphi$ values range from -1 to 1, with values closer to 1 indicating a stronger positive correlation. Cohen's $\kappa$ represents the agreement between classifications beyond chance, whereas $r_\varphi$ quantifies the strength of association between classification methods. Fifth, we investigated group differences regarding the three classification methods with gender, age, depressive and social anxiety symptoms, using $\chi^2$-tests for categorical scales and analysis of variance or $t$-tests for the continuous scales. Sixth, we performed three (for each classification method) two-sided ANCOVA to test group differences regarding interpretation bias, rejection sensitivity, social avoidance behavior, distress disclosure, self-esteem, and avoidance goal intensity while controlling for depressive (PHQ-9) and social anxiety symptoms (SIAS-6) by including them as continuous covariates. This approach allowed us to isolate the specific effects of loneliness facets on cognitive and behavioral variables and ensured that observed differences are not solely driven by underlying psychopathology. Finally, we calculated partial $\eta2$ for the effect sizes. To account for false discovery rate due to multiple testing, we implemented the Benjamini-Hochberg correction method [73] for the ANCOVAs. The Benjamini-Hochberg correction controls the false discovery rate by ranking p-values in ascending order and adjusting them based on their rank and the total number of tests.

## 3. Results

### 3.1. Descriptive statistics and zero-order correlations

Means, standard deviations, medians, ranges, and zero-order correlations of all study variables, as well as Cronbach's alpha of all measures, are presented in Table 1. The zero-order correlations revealed that all three facets of loneliness showed significant associations with all constructs connected to the cognitive model of loneliness. The associations of loneliness frequency, loneliness distress, and loneliness chronicity with interpretations bias, rejection sensitivity, social avoidance behavior, and avoidance goal intensity were significantly positive and negative with distress disclosure and self-esteem.

### 3.2. Distribution of lonely vs. non-lonely and group characteristics

In our sample, based on the different approaches, either 15.95% (frequency), 29.75% (distress), or 19.49% (chronicity) were classified in the lonely group compared to the non-lonely group, in all classification methods, lonely and non-lonely people significantly differed regarding age, depressive and social anxiety symptoms (see Table 2).

Across all approaches, 65.06% of the participants were classified as not lonely in all three aspects, while 12.03% were classified as lonely in one, 15.57% in two, and 7.34% in all three aspect(s).

### 3.3. Degree of agreement between classification

Overall, the $\chi^2$-analyses revealed significant differences regarding the distribution (see Table 3). The percentage agreement (see Fig 1) between the loneliness *frequency* and loneliness *distress* classification was 78.10%, with a Cohen's $\kappa$ of .40, indicating almost moderate agreement; with $r_\varphi$ =.43, suggesting a strong positive association. For the loneliness *frequency* and loneliness *chronicity* classification, the percentage agreement was 79.74%, with a Cohen's $\kappa$ of .31, indicating fair agreement; and $r_\varphi$ =.31, suggesting a moderate positive association. The comparison between the loneliness

**Table 2. Distribution of lonely and non-lonely participants within each approach.**

| | Loneliness Frequency | | Group Difference |
|---|---|---|---|
| | Lonely | Not Lonely | |
| **N** | 126 (15.95%) | 664 (84.05%) | |
| **Gender (women in %)** | 79.37% | 81.33% | $\chi^2(1) = 0.08$ |
| **Mean Age (SD)** | 35.25 (14.24) | 31.19 (12.02) | $t(160.53) = -3.00**$ |
| **Depressive Symptoms (SD)** | 15.16 (5.93) | 9.29 (5.81) | $t(173.64) = -10.22***$ |
| **Social Anxiety Symptoms (SD)** | 10.48 (5.66) | 5.58 (4.59) | $t(157.73) = -9.16***$ |
| | **Loneliness Distress** | | |
| | Lonely | Not Lonely | |
| **N** | 235 (29.75%) | 555 (70.25%) | |
| **Gender (women in %)** | 79.15% | 81.80% | $\chi^2(1) = 0.58$ |
| **Mean Age (SD)** | 33.69 (13.36) | 31.05 (12.02) | $t(402.01) = -2.61**$ |
| **Depressive Symptoms (SD)** | 15.23 (5.79) | 8.10 (5.07) | $t(393.15) = -16.40***$ |
| **Social Anxiety Symptoms (SD)** | 9.24 (5.50) | 5.14 (4.38) | $t(365.75) = -10.15***$ |
| | **Loneliness Chronicity** | | |
| | Lonely | Not Lonely | |
| **N** | 154 (19.49%) | 636 (80.51%) | |
| **Gender (women in %)** | 75.97% | 82.23% | $\chi^2(1) = 3.18$ |
| **Mean Age (SD)** | 34.30 (12.96) | 31.24 (12.30) | $t(224.44) = -2.65**$ |
| **Depressive Symptoms (SD)** | 15.23 (5.88) | 8.81 (5.66) | $t(226.62) = -11.87***$ |
| **Social Anxiety Symptoms (SD)** | 9.56 (5.63) | 5.59 (4.64) | $t(206.33) = -8.12***$ |

Notes.

*$p <.05$,

**$p < 0.01$,

***$p <.001$. $N = 790$.

distress and loneliness chronicity classification showed an 86.96% agreement, with a Cohen's $\kappa$ of .65, indicating substantial agreement, and $r_\varphi$ =.68, indicating a strong positive relationship. In sum, these results indicate that while there is some overlap between different loneliness classification methods, they are not entirely interchangeable. The varying levels of agreement across measures suggest that each method captures a distinct aspect of loneliness, with distress and chronicity showing the highest alignment and frequency showing lower agreement with the other methods.

Table 3 displays the comparison of the three methods of classification with each other. As for the comparison between frequency and distress, most individuals who were not classified lonely in terms of distress also did not in terms of frequency (94.23%). A small fraction of individuals who were not classified as lonely in terms distress nonetheless were classified as lonely in terms of frequency (5.77%). Over half of the individuals who were classified as lonely in terms of distress were not classified as such in terms of frequency (60.00%). Nearly half of the individuals who classified as lonely in terms of distress also classified as such in terms of frequency (40.00%). Regarding the comparison between frequency and chronicity, most individuals who were not classified as lonely in terms of chronicity were also not classified as lonely in terms of frequency (89.62%). A smaller portion of individuals who were not classified as lonely in terms of chronicity were nonetheless classified as lonely in terms of frequency (10.38%). Over half of the individuals who were classified as lonely in terms of chronicity were not classified as such in terms of frequency (61.04%). Nearly half of the individuals who were classified as lonely in terms of chronicity were also classified as such in terms of frequency (38.96%). Finally, the comparison between distress and chronicity revealed that

**Table 3. Comparison of the three methods of classification.**

| | | Frequency | | |
| --- | --- | --- | --- | --- |
| | | **Not Lonely** | **Lonely** | |
| **Distress** | **Not Lonely** | 523 (94.23%) | 32 (5.77%) | 555 |
| | **Lonely** | 141 (60.00%) | 94 (40.00%) | 235 |
| | | 664 | 126 | 790 |
| | | Frequency | | |
| | | **Not Lonely** | **Lonely** | |
| **Chronicity** | **Not Lonely** | 570 (89.62%) | 66 (10.38%) | 636 |
| | **Lonely** | 94 (61.04%) | 60 (38.96%) | 154 |
| | | 664 | 126 | 790 |
| | | Chronicity | | |
| | | **Not Lonely** | **Lonely** | |
| **Distress** | **Not Lonely** | 544 (98.02%) | 11 (1.98%) | 555 |
| | **Lonely** | 92 (39.15%) | 143 (60.90%) | 235 |
| | | 636 | 154 | 790 |

*Notes.* Frequency and Distress: $\chi^2(1) = 147.79$, $p < .001$; Frequency and Chronicity: $\chi^2(1) = 73.45$, $p < .001$; Distress and Chronicity: $\chi^2(1) = 360.83$, $p < .001$.

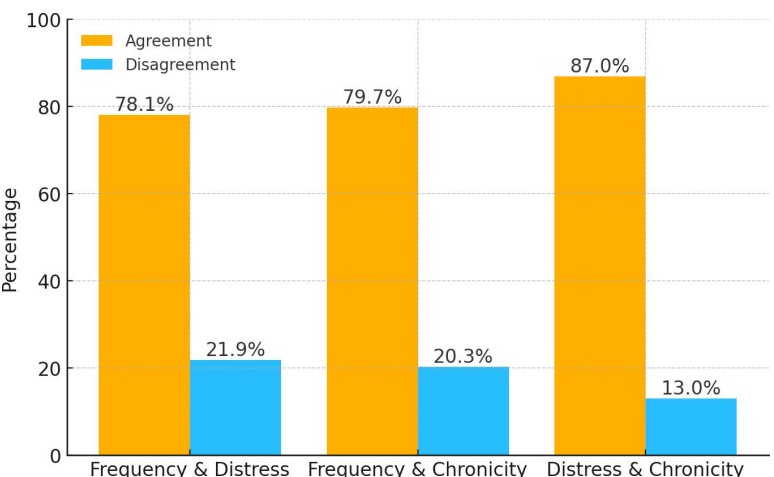

**Fig 1. Agreement and disagreement between loneliness measures.**

a vast majority of individuals who were not classified as lonely in terms of distress were also not classified as lonely in terms of chronicity (98.02%). A very small proportion of individuals who were not classified as lonely in terms of distress were nonetheless classified as lonely in terms of chronicity (1.98%). More than a third of individuals who were classified as lonely in terms of distress were not classified as such in terms of chronicity (39.10%). Most individuals who were classified as lonely in terms of distress were also classified as such in terms of chronicity (60.90%).

### 3.4. Group differences regarding cognitive and behavioral aspects

Table 4 presents the results of the ANCOVAs testing the mean differences between lonely vs. non-lonely groups for the dependent variables interpretation bias, rejection sensitivity, social avoidance behavior, distress disclosure, self-esteem,

**Table 4. Mean difference between lonely and non-lonely people regarding cognitive and behavioral aspects.**

| Dependent Variables | Loneliness Frequency | | | |
| --- | --- | --- | --- | --- |
| | M (SD) | | F | partial η2 |
| | Lonely | Not Lonely | | [95% CI] |
| **Interpretation Bias** | 2.29 (.69) | 1.76 (.55) | 111.80*** | .12 [.09,.17] |
| **Rejection Sensitivity** | 14.11 (5.74) | 8.80 (3,93) | 215.19*** | .22 [.17,.27] |
| **Social Avoidance Behavior** | 24.25 (7.10) | 18.68 (6.28) | 122.00*** | .14 [.09,.18] |
| **Distress Disclosure** | 32.75 (11.42) | 38.34 (9.81) | 47.55*** | .06 [.03,.09] |
| **Self-Esteem** | 12.36 (6.70) | 19.53 (6.57) | 227.11*** | .22 [.18,.27] |
| **Avoidance Goal Intensity** | 3,83 (.62) | 3.62 (.53) | 15.74*** | .02 [.01,.04] |
| | Loneliness Distress | | | |
| | M (SD) | | F | partial η2 |
| | Lonely | Not Lonely | | [95% CI] |
| **Interpretation Bias** | 2.14 (.68) | 1.72 (.53) | 105.41*** | .12 [.08,.16] |
| **Rejection Sensitivity** | 12.73 (5.21) | 8.34 (3.76) | 224.95*** | .23 [.18,.28] |
| **Social Avoidance Behavior** | 22.51 (6.98) | 18.32 (6.22) | 106.88*** | .12 [.08,.16] |
| **Distress Disclosure** | 35.62 (10.33) | 38.42 (10.17) | 19.02*** | .03 [.01.03] |
| **Self-Esteem** | 13.24 (6.08) | 20.57 (6.33) | 370.39*** | .32 [.27,.37] |
| **Avoidance Goal Intensity** | 3.91 (.49) | 3.55 (.54) | 77.43*** | .09 [.06,.13] |
| | Loneliness Chronicity | | | |
| | M (SD) | | F | partial η2 |
| | Lonely | Not Lonely | | [95% CI] |
| **Interpretation Bias** | 2.18 (.71) | 1.76 (.55) | 81.53*** | .09 [.06,.13] |
| **Rejection Sensitivity** | 12.89 (5.55) | 8.86 (4.09) | 139.58*** | .16 [.11,.20] |
| **Social Avoidance Behavior** | 23.34 (6.78) | 18.65 (6.40) | 126.91*** | .11 [.08,.16] |
| **Distress Disclosure** | 35.43 (10.52) | 38.99 (10.21) | 15.94*** | .02 [.01,.04] |
| **Self-Esteem** | 13.06 (6.02) | 19.68 (6.73) | 224.00*** | .22 [.17,.27] |
| **Avoidance Goal Intensity** | 3.88 (.50) | 3.60 (.55) | 35.49*** | .04 [.02,.07] |

Notes. Age, Depressive and Social Anxiety Symptoms were implemented as covariates. Partial η² values indicate the proportion of variance explained by loneliness classification in each cognitive-behavioral variable. Values below.01 suggest a negligible effect,.01-.06 a small effect,.06-.14 a moderate effect, and above.14 a large effect.

*p <.05,

**p < 0.01,

***p <.001 N = 790.

and avoidance goal intensity, with age, depressive and social anxiety symptoms as covariates, including partial η² for the effect sizes.

Regarding the groups based on loneliness frequency, the ANCOVA results indicated significant main effects of loneliness score regarding all dependent variables (p-values <.001). They indicate that lonely vs. non-lonely people, in regard to frequency, report more interpretation bias (moderate effect), higher rejection sensitivity (large effect), more social avoidance behavior (moderate effect), lower distress disclosure (medium effect), and lower self-esteem (large effect), as well as increased avoidance goal intensity (small effect).

Similarly, significant main effects were also found in all dependent variables for loneliness distress. These findings indicate that higher distress associated with loneliness is linked to greater interpretation bias (medium effect), higher rejection sensitivity (large effect), more social avoidance behavior (medium effect), lower distress disclosure (small effect), and lower self-esteem (large effect), as well as increased avoidance goal intensity (small effect).

Significant effects of chronicity were found in all dependent variables for loneliness chronicity. Significant group differences between lonely and non-lonely individuals in terms of chronicity were found in interpretation bias (medium effect), higher rejection sensitivity (large effect), more social avoidance behavior (medium effect), lower distress disclosure (medium effect), and lower self-esteem (large effect), as well as higher avoidance goal intensity (small effect).

## 4. Discussion

The present study aimed to explore the cognitive and behavioral differences between lonely and non-lonely individuals by incorporating three different aspects of loneliness. Specifically, we built groups of lonely vs. non-lonely people based on frequency, distress, and chronicity. We then tested the degree of agreement between the three classification methods before comparing lonely vs. non-lonely individuals regarding cognitive and behavioral aspects within all classifications.

### 4.1. Lonely vs. non-lonely individuals based on frequency, distress, and chronicity

The three methods of group building revealed different distributions regarding who is considered lonely and who is not. More participants were classified as lonely with the direct single item connected to distress (29.8%) than with the indirect measure assessing loneliness frequency (16.0%). This stands in line with the findings by Shiovitz-Ezra and Ayalon [34], who also found higher prevalences of lonely individuals when asked directly. However, this contrasts with most previous studies, which either showed similar distributions [e.g., 36] or higher when assessed indirectly [e.g., 74]. Furthermore, this also contradicts findings pointing toward a self-stigma connected with feelings of loneliness [75–77]. As for the degree of agreement between the loneliness frequency and distress method, the results reveal almost moderate to strong agreement. In comparison to the findings of Shiovitz-Ezra and Ayalon [34], the overlap of participants being classified as lonely in both terms was larger. However, over half of the individuals (60%) who were classified in terms of distress (direct measure) were not classified as lonely in terms of frequency (indirect measure), which is in line with the findings by Shiovitz-Ezra and Ayalon [34]. The discrepancies regarding the different distributions and degree of agreement with these findings might emerge due to the possibility that when asked directly, individuals may be more likely to acknowledge and report these feelings due to the immediate reflection on their emotional state. This could contrast with indirect measures that assess loneliness more broadly and abstractly, possibly diluting the immediate emotional impact and leading to lower reported levels of loneliness. Another possibility could be that by using the word "loneliness" in direct measures, individuals can use their own interpretation of loneliness when reporting such feelings. Indirect measures might not account for some of these interpretations, which could mean that certain items do not reflect aspects of an individual's concept of loneliness. Taken together, our findings suggest that while there is some degree of overlap between the two classification methods, each one of them seems to capture unique aspects of loneliness and, therefore, leads to a different distribution in the two groups.

Our classification methods for loneliness frequency and loneliness duration revealed distinct distributions, with a higher percentage of participants classified as lonely when considering chronicity (19.5%) compared to frequency (16.0%). More than half of the participants who were classified as lonely in terms of chronicity were not classified as lonely in terms of frequency (61.0%). This might suggest that some aspects of longer periods of loneliness are not always observable through indirect methods. The fair to moderate degree of agreement also suggests that while there is an overlap, the two methods might diverge due to unique aspects of loneliness.

Our findings indicate that a lower percentage of participants were classified as lonely when assessed for chronicity (19.5%) compared to distress (29.8%). Around 40% of participants classified as lonely in terms of distress did not classify as lonely for longer than two years. However, the other 60% of the participants who were classified as lonely in terms of chronicity were not classified as such in terms of distress. This is also reflected by a substantial agreement. While the immediacy and salience of emotional distress might lead to higher reporting rates of loneliness distress due to the direct impact on an individual's well-being, most of the participants experiencing loneliness for longer durations also feel the

distress connected to loneliness. However, it has to be noted that the overlap might also partially emerge due to the direct single item being involved in the operationalizing of both loneliness distress and chronicity.

Taken together, our analysis of the degree of agreement between the three distinct classification methods of loneliness - frequency, distress, and chronicity - illuminate some overlaps and important divergences that reflect the complex nature of loneliness as a multifaceted construct. The fair to substantial agreement observed between these methods underscores their relative reliability but also highlights the nuances that each dimension captures about the experience of loneliness. This and the fact that some participants have been classified as lonely in none, one or several aspects of loneliness supports this notion. However, more research is needed to grasp and fully entangle the nature of loneliness and its different aspects [11].

## 4.2. Further support for distorted social information process in lonely people

Comparing lonely and non-lonely individuals indicated significant differences in all investigated cognitive, behavioral and motivational aspects regardless of the classification (frequency, distress, and chronicity), even after controlling for age, depressive and social anxiety symptoms. Higher levels of loneliness were associated with greater interpretation bias, rejection sensitivity, social avoidance behavior, lower distress disclosure, and self-esteem, as well as increased avoidance goal intensity. These findings are consistent with the cognitive model of loneliness proposed by Caciopppo and Hawkley [12] and the findings in the review by Spithoven et al. [13].

Lonely individuals exhibited significantly greater interpretation bias, aligning with previous research demonstrating that they tend to interpret ambiguous social situations negatively [15,16]. This bias, in turn, can reinforce rejection sensitivity, where we found significant group differences. Lonely individuals tend to anticipate rejection, leading them to be more vigilant for social threats [78,79], which is theorized to promote social withdrawal [10,12]. Accordingly, our results show that loneliness comes with increased social avoidance behavior. This finding is in line with studies showing that lonely individuals are more likely to engage in behaviors that avoid social interactions [79,80]. Notably, rejection sensitivity has been shown to predict social avoidance behavior [79] and thus may potentially exacerbate feelings of loneliness. These tendencies may manifest in everyday social settings, such as hesitancy to initiate conversations in group settings, reluctance to seek emotional support, or misinterpreting neutral social cues as rejection. Over time, such patterns could contribute to difficulties in maintaining friendships and hinder the formation of new social bonds.

Additionally, our results suggest that loneliness appears to be linked to higher avoidance goal intensity, which stands in line with previous research [81,82]. This means that lonely individuals seem more motivated to avoid negative outcomes. Similarly, Mund and Neyer [83] found significant associations between loneliness and prevention focus (i.e., avoiding harm in social situations). This increased avoidance motivation might contribute to reduced distress disclosure, which corroborates findings of previous studies [84,85] and could potentially hinder relationship formation and deepen (subjective) isolation. Finally, the large effect sizes of the group differences observed for self-esteem suggest that self-esteem is a central feature of all three facets of loneliness. Previous research has consistently found a strong link between loneliness and self-esteem [20,21,86], indicating that low self-esteem might reinforce loneliness. However, there is also the potential for high self-esteem having a buffering effect on loneliness.

While our findings highlight significant cognitive and behavioral differences between lonely and non-lonely individuals, future research could explore potential mediating or moderating mechanisms that shape these relationships. For instance, cognitive biases, such as negative interpretation bias, may mediate the link between loneliness and social withdrawal, reinforcing the cycle of loneliness, as previous cross-sectional studies have suggested [22]. Similarly, self-esteem could serve as a protective factor, potentially buffering against the negative effects of loneliness. Prior research has shown that self-esteem moderates the impact of loneliness on life satisfaction and symptoms of depression and anxiety [87–90]. Examining these mechanisms could provide deeper insights into how loneliness develops and persists, ultimately informing more targeted interventions.

In sum, our analyses confirm the substantial relationship of loneliness with various cognitive and behavioral variables, consistent with the findings by Spithoven et al. [13]. Future research should test how these different components of the cognitive model of loneliness might affect each other over time.

### 4.3. Maladaptive vs. adaptive loneliness

As we circle back to the initial discussion on defining maladaptive loneliness, our results invite reconsidering whether a singular facet of loneliness can fully encapsulate its maladaptive nature. Our findings suggest that different aspects of loneliness, i.e., frequency, distress and chronicity, are associated with maladaptive cognitive, behavioural, and motivational aspects, providing further support for a multifaceted approach [11,31]. Implementing such an approach might provide distinct profiles, which could lead to a clearer prediction of maladaptive loneliness. Similar to Maes and Vanhalst [11], we argue that the inclusion of additional cognitive and behavioral constructs might further help conceptualize maladaptive loneliness since theoretical frameworks suggest the intertwined relationship between them [e.g., 10,12]. However, a major remaining open questions is the scarce research regarding the adaptive nature of loneliness [see for review: 11]. A possible reason for the absence of adaptive tendencies in our results could be the cross-sectional design and the focus on more trait-like loneliness, which does not allow for the examination of situational processes that might be adaptive, even if the overall pattern of loneliness appears maladaptive. For instance, Reissmann et al. [8] used experience sampling methodology (ESM) to track fluctuations in social interactions in response to momentary feelings of loneliness, prompting participants multiple times a day to report their loneliness levels and social interactions. Their findings indicated that loneliness led to both increases and decreases in subsequent social interactions [8].

### 4.4. Implications

Our study underscores the complex and multifaceted nature of loneliness, demonstrating that it encompasses more than mere frequency. This reinforces the need to consider more aspects of loneliness - such as emotional distress and chronicity - when studying its impacts and developing interventions. Recognizing these diverse dimensions can refine theoretical models and enhance intervention specificity. Further, the development of comprehensive measurement tools that capture the multifaceted nature of loneliness is crucial [30,31]. These tools should be validated across different cultures and demographics to ensure their sensitivity and accuracy. In practice, our findings suggest that interventions should be tailored, addressing the emotional distress and the cognitive and behavioral patterns associated with loneliness and/or enhancing social skills and/or self-esteem [91,92]. Similarly, Käll et al. [39] argue against "one-size fits all" approaches and suggest modular interventions that address specific cognitive, emotional, and behavioral mechanisms of loneliness may be more effective. For instance, those experiencing frequent loneliness might benefit from cognitive restructuring, while individuals with high distress could focus on emotional regulation, and those with chronic loneliness may require interventions targeting long-term avoidance patterns and social skills training. However, future research is needed to systematically examine how different intervention strategies can be tailored to distinct dimensions of loneliness. Overall, this study highlights the importance of a comprehensive understanding and targeted interventions to effectively address loneliness, contributing to improved mental health outcomes and social connections for those affected.

### 4.5. Limitations

Despite the strengths of this study, several limitations must be acknowledged. First, while our findings indicate significant differences in cognitive and behavioral processes across different loneliness classifications, the cross-sectional design does not allow for conclusions about causality. Future research should employ longitudinal designs to disentangle whether cognitive biases and behavioral tendencies contribute to loneliness persistence or emerge as a consequence of prolonged loneliness. Second, a potential limitation of this study is the presence of self-selection bias, which may have

influenced the characteristics of our sample. Given that our study was explicitly advertised as a study on loneliness, it is possible that individuals who experience loneliness were more motivated to participate, leading to an overrepresentation of lonely individuals. This may have inflated prevalence rates. Third, the sample was predominantly female (81%), which limits generalizability. While a previous meta-analysis has found that gender differences in loneliness are generally small [93], a newer study reported higher levels of loneliness in men than women [94]. Fourth, our sample was relatively young. While this limits the generalizability of our findings to older or adolescent populations, prior research suggests that key cognitive and behavioral mechanisms underlying loneliness - such as interpretation bias and social avoidance - are relevant across age groups [10,13]. Nonetheless, future studies should investigate whether the relationships observed in our study differ across developmental stages. Fifth, the reliance on self-report measures introduces potential response biases. Participants may have underreported or overreported their feelings due to social desirability or recall bias. Incorporating objective measures of behavioral tendencies [e.g., 95] and cognitive biases [e.g., 96] could help mitigate these biases. Sixth, we also did not account for all potential confounding variables. While controlling for depressive and social anxiety symptoms, factors like personality traits, quality of interpersonal relationships, and recent life events were not considered. Including these factors in future studies would strengthen the findings. Seventh, our operationalization of loneliness chronicity as feelings lasting at least 24 months, while grounded in previous literature, may not fully capture the nuanced experiences of chronic loneliness. More refined measures of chronicity in longitudinal studies could provide deeper insights. Eighth, even though the cut-off points were extracted from previous studies or theories, they remain somewhat arbitrary. Changing these would have also changed the results. Nevertheless, the three methods did not seem to capture the same individuals entirely. Ninth, this study's assessment of chronic loneliness covered the past 24 months, a period that overlapped with the COVID-19 pandemic. This raises the possibility that some participants may have perceived the duration of their loneliness as chronic due to social distancing and lockdowns, even if it could have been primarily situational. However, research suggests that while overall loneliness increased during the pandemic, the effects were small [6]. Given that loneliness is influenced by stable cognitive and emotional factors rather than just external circumstances, the extent of this bias remains unclear. Future research should compare post-pandemic data to determine whether chronic loneliness rates have returned to pre-pandemic levels. Finally, this study it should be mentioned that this study was not pre-registered, which limits transparency in hypothesis testing.

## 5. Conclusion

In conclusion, this study provides further support that loneliness is associated with significant cognitive, behavioral and motivational maladaptations regardless of the classification method. These findings underscore the importance of investigating and addressing loneliness as a multifaceted construct and suggest that interventions should focus on cognitive-behavioral components to mitigate its negative effects. Further research, particularly longitudinal studies, is needed to build on these findings, to further investigate the distinction between adaptive and maladaptive loneliness and develop effective strategies to combat loneliness.

## Acknowledgments

We thank Milena Imwinkelried and Alessandra Stanizzi for their help with data collection. Further, we thank Johanna Boettcher for providing us with the German version of the Interpretation and Judgmental Bias Questionnaire.

## Author contributions

**Conceptualization:** Andrej Skoko, Tobias Krieger.

**Formal analysis:** Andrej Skoko.

**Funding acquisition:** Tobias Krieger.

**Investigation:** Andrej Skoko, Noëmi Seewer, Tobias Krieger.

**Methodology:** Andrej Skoko.

**Supervision:** Tobias Krieger.

**Writing – original draft:** Andrej Skoko.

**Writing – review & editing:** Noëmi Seewer, Marcus Mund, Tobias Krieger.

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
