## [Decision Letter · Decision Letter 0]

4 Nov 2024

PONE-D-24-42587Revisiting the Cognitive and Behavioral Aspects of Loneliness: Insights from Different Measurement ApproachesPLOS ONE

Dear Dr. Skoko,

Thank you for submitting your manuscript to PLOS ONE. After careful consideration, we feel that it has merit but does not fully meet PLOS ONE’s publication criteria as it currently stands. Therefore, we invite you to submit a revised version of the manuscript that addresses the points raised during the review process. 

We look forward to receiving your revised manuscript.

Kind regards,

Bao-Liang Zhong

Academic Editor

PLOS ONE

Journal Requirements:

Reviewers' comments:

Reviewer's Responses to Questions

**Comments to the Author**

1. Is the manuscript technically sound, and do the data support the conclusions?

Reviewer #1: Yes

Reviewer #2: Yes

Reviewer #3: Yes

Reviewer #4: Yes

Reviewer #5: Partly

2. Has the statistical analysis been performed appropriately and rigorously?

Reviewer #1: Yes

Reviewer #2: Yes

Reviewer #3: Yes

Reviewer #4: Yes

Reviewer #5: Yes

3. Have the authors made all data underlying the findings in their manuscript fully available?

Reviewer #1: Yes

Reviewer #2: Yes

Reviewer #3: Yes

Reviewer #4: Yes

Reviewer #5: Yes

4. Is the manuscript presented in an intelligible fashion and written in standard English?

Reviewer #1: Yes

Reviewer #2: Yes

Reviewer #3: Yes

Reviewer #4: Yes

Reviewer #5: Yes

5. Review Comments to the Author

Reviewer #1: I would like to compliment the authors on a really well written and clear paper. It doesn't promise anything it can't deliver, and I recognize that the authors explicitly point out its exploratory nature. I would encourage the authors to also upload their analysis scripts to OSF and not just the raw data - I would have liked to have had a look at the code. I think the paper will be a great contribution to the literature, and I have only relatively minor suggestions before I can recommend acceptance of the paper.

Abstract

1. This is just my personal opinion, but “ancova” as a keyword seems strange to me. It's not a specific modeling approach that people might be looking for, nor is it a content-related keyword.

Introduction

2. In the research I was involved in, we looked at the fact that loneliness should be differentiated at the state and trait level. This aspect could fit well in the introduction (only if the authors consider it relevant).

Gründahl, M., Weiß, M., Maier, L., Hewig, J., Deckert, J., & Hein, G. (2022). Construction and validation of a scale to measure loneliness and isolation during social distancing and its effect on mental health. Frontiers Psychiatry, 13, 798596.

3. Overall, I really liked the introduction very much. If the authors could think about how to tighten it up a bit, I think a lot of people might enjoy reading it. The rationale for the approach is well explained. It's a bit sad that the authors did not pre-register their study.

Methods

4. Please provide further information on the survey: How many participants dropped out of the study? Was there a data-related quality/exclusion criterion (e.g., straightlining)? Were the questionnaires presented in random order or always in the same order (which order?)?

Results

5. As the authors state in their introduction, loneliness can also maintain or further reinforce maladaptive cognitive and behavioral patterns. Since this study is cross-sectional, the direction is not clear. I wondered if adding three logistic regressions (one for each division method) using all the outcomes from the ancovas as predictors could also account for this aspect? This is by no means mandatory, it was just an idea that could also be argued against.

Limitations

6. I don't want to start an unrelated Covid debate here. But the question about the last 24 months in 2021/2022 could have led to a distorted self-assessment of chronic loneliness, as many people may have thought about lockdowns etc. This could also (partly) explain the relatively high prevalence of chronic loneliness.

Reviewer #2: I would like to express my sincere gratitude for the opportunity to review your manuscript titled "Revisiting the Cognitive and Behavioral Aspects of Loneliness: Insights from Different Measurement Approaches." I deeply appreciate the meticulous and rigorous work you have conducted, and the significant contribution it makes to understanding the cognitive and behavioral models related to loneliness.

Your study highlights the complexity of loneliness and explores its various dimensions, providing a more nuanced and detailed view of this phenomenon, thanks to the use of distinct measures that account for frequency, duration, and distress. The results, which show significant differences in all the cognitive and behavioral components analyzed, form a solid basis for targeted interventions that can more precisely address different forms of loneliness.

After careful review, I would like to offer the following suggestions to further improve the manuscript:

Lines 30-32: The statement on the division of the sample into "lonely and not lonely" individuals could be clarified. I suggest adding a more detailed description of the methodology used to distinguish between the different groups, specifying the criteria for each measure more clearly.

Lines 49-51: The control of depressive and social anxiety symptoms is mentioned. It would be helpful to expand this section by providing more details on the methods and tools used to control for these factors.

Lines 121-125: The definition of "chronic loneliness" could benefit from further elaboration, explaining the choice of the 24-month period as a threshold for chronicity, perhaps integrating additional studies that support this choice.

Lines 206-210: The results regarding the agreement between the three loneliness measures (frequency, distress, chronicity) could be supported by explanatory graphs that visually illustrate the percentage of agreement and disagreement between the different measures.

Lines 255-265: The interpretation of the correlation between loneliness and social avoidance behavior could be enriched by discussing possible psychological mechanisms that might explain this link.

Lines 350-355: I suggest a broader discussion of the study's limitations. Although depressive and social anxiety symptoms were included, there may be other confounding factors, such as social networks or the quality of interpersonal relationships, that deserve to be mentioned as potential influences not accounted for.

Lines 400-405: It would be helpful to include a dedicated section on the study’s limitations. For example, the cross-sectional nature of the study design might limit causal conclusions. Additionally, the influence of potential self-selection bias in the sample collection should be considered, given that most participants were women and the average age was relatively young.

Lines 425-430: Lastly, I recommend expanding the clinical implications section by providing more concrete examples of how interventions can be adapted to the different dimensions of loneliness identified.

Regarding the data analysis and results, the statistical methods used are appropriate and rigorous, with suitable controls for depressive and social anxiety symptoms. However, I would recommend providing more details on the adjusted p values using the Benjamini-Hochberg method and on interaction effects between the cognitive and behavioral variables. No specific sections need to be eliminated, but I suggest streamlining some parts of the discussion to reduce repetition and make the text more fluid.

In conclusion, the work presented is of great scientific value and significantly contributes to the existing literature. I hope these suggestions will help further refine the manuscript. Once again, thank you for entrusting me with the review of this important contribution.

I would kindly suggest that you include a reference to the following article in your manuscript:

Diotaiuti, P., Valente, G., Mancone, S., Grambone, A., & Chirico, A. (2021). Metric goodness and measurement invariance of the Italian brief version of Interpersonal Reactivity Index: A study with young adults. Frontiers in Psychology, 12, Article 773363. https://doi.org/10.3389/fpsyg.2021.773363.

This article is highly relevant for your discussion on measurement tools, particularly when addressing the methodological aspects related to the validation and reliability of psychological scales. I recommend citing this work in the section where you discuss the psychometric properties of the tools used to measure loneliness, specifically around Lines 150-160, where you explore the reliability and validity of different scales. The reference could enrich your discussion on the robustness of measurement instruments across different cultural and demographic groups.

Reviewer #3: The study "Revisiting the Cognitive and Behavioral Aspects of Loneliness: Insights from Different Measurement Approaches" conducted a comprehensive examination of the cognitive and behavioral differences associated with various facets of loneliness in a sample of 790 German-speaking adults. The researchers employed three distinct measures of loneliness—frequency, distress, and chronicity—to categorize participants as lonely or not lonely. They then analyzed group differences in cognitive and behavioral aspects such as interpretation bias, social avoidance, self-esteem, and rejection sensitivity, while controlling for depressive and social anxiety symptoms. The findings revealed fair to substantial agreement between the three loneliness measures and significant group differences in all cognitive and behavioral components for each loneliness measure. This highlights the multifaceted nature of loneliness and the importance of using diverse measures to capture its complexity. The study contributes to a more nuanced understanding of loneliness and suggests that interventions should consider specific dimensions of loneliness to effectively address its cognitive and behavioral implications. I suggest the authors to directly indicate the three definitions of loneliness and the study participants in the title, to be strict. In the introduction, when describing the negative mental and physical health outcomes of enduring loneliness, the authors must be aware of the limited availability of evidence on some consequences of persistent loneliness (PMID: 24550354, PMID: 26905049). I suggest the authors to revise the sentence “While nearly everyone experiences loneliness at some point in their life, for some, it can become an enduring condition with significant negative implications for mental and physical health and even increased mortality, elevating loneliness as a global health priority”. In the part of assessing loneliness, please consider to review the De Jong-Gierveld Loneliness Scale and explain why the authors did not use it. In the methodology, please consider the poor representativeness of the study sample. The authors need to be aware of the different constructs of loneliness in adolescents, adults, and older adults. The online survey was conducted during the COVID-19 pandemic but the authors did not have comments on this special social context (PMID: 37562972). My concern the generalizability of the current findings.

Reviewer #4: General Assessment:

The manuscript presents a well-designed and timely study exploring the multifaceted nature of loneliness, using three distinct measures: loneliness frequency, distress, and chronicity. The study's emphasis on understanding the cognitive and behavioral aspects of loneliness through diverse measurement approaches is commendable and contributes to the ongoing discourse on the complexity of loneliness. I do have some comments which I hope will help to authors to further strengthen the manuscript:

1. While the authors acknowledge the cross-sectional nature of the study, additional discussion on how self-selection bias might have impacted the findings (given that participants self-identified for a loneliness study) would be beneficial. More emphasis on potential biases due to the high proportion of female participants (81%) and the relatively young sample would also strengthen the paper.

2. The use of a self-reported measure of chronicity (24 months) could be refined further. While grounded in previous literature, it may not fully capture the nuanced experiences of chronic loneliness, and a more detailed discussion of why this specific threshold was used and its potential limitations would add clarity.

3. Although the manuscript provides a solid analysis of the cognitive and behavioral components of loneliness, the reliance on self-report questionnaires introduces possible response biases (e.g., social desirability or recall biases). Addressing the potential limitations of self-reported data and suggesting alternative or supplementary measures (such as behavioral data) would provide more balance.

4. The implications for interventions could be expanded. While the discussion touches on how different facets of loneliness could guide targeted interventions, more specific examples or suggestions (e.g., interventions tailored to chronic vs. distress-related loneliness) would enhance the practical applicability of the findings.

Specific Comments:

Abstract: The abstract is succinct and informative but could benefit from specifying the sample characteristics (age range, gender distribution) to provide context for the generalizability of the findings.

Introduction: The introduction could briefly address the global health implications of loneliness, especially its rising importance during post-pandemic times, which would make the study even more relevant.

Results: The presentation of statistical results is thorough. However, some readers might benefit from a more intuitive explanation of Cohen's κ and rϕ values, particularly in the context of agreement between the loneliness measures.

Tables: The tables are well-organized and provide essential data. Adding brief interpretative captions below some of the more complex tables (e.g., explaining what constitutes a “strong positive association” in lay terms) could make the findings more accessible to a broader readership.

Reviewer #5: This study presents valuable insights into the cognitive and behavioral aspects of loneliness. The topic of loneliness is highly relevant, particularly in the context of contemporary societal issues related to mental health, the authors provide a comprehensive review of existing literature, framing their research within established theoretical frameworks, and the study employs a multi-dimensional approach to assess loneliness through frequency, distress, and chronicity, which is commendable. However, it requires a more critical approach to theoretical frameworks, measurement choices, and the interpretation of results. Addressing these issues would enhance the overall rigor and impact of the research. The specific issues are as follows.

1. Clarification of theoretical frameworks: the manuscript heavily leans on established theoretical frameworks. However, it lacks a critical assessment of these frameworks. For instance, while the text acknowledges the adaptive aspects of loneliness, it does not thoroughly explore how this adaptive nature contrasts with the maladaptive outcomes. furthermore, a more nuanced discussion regarding the interplay between adaptive and maladaptive loneliness would enhance the theoretical grounding of the study.

2. Measurement choices: the authors critique the use of the UCLA Loneliness Scale and other measurement tools for their limitations, such as not capturing the intensity or duration of loneliness adequately. However, the manuscript does not provide a clear rationale for the selected measures or how they were specifically adapted for this study. In addition, the choice of cut-off points for categorizing loneliness is somewhat arbitrary and lacks a robust justification.

3. Sample limitation: the sample is predominantly female (81%) and relatively young (mean age of 31.86 years). While the authors note the limitations of generalizability, they do not adequately explore how these demographic factors might influence the findings. For example, the experiences and expressions of loneliness may differ significantly across genders and age groups.

4. While the statistical analyses employed, such as ANCOVA and χ2 tests, are appropriate, the manuscript could benefit from a more detailed explanation of why these specific methods were chosen over others. Additionally, the analyses focus on group differences, but they lack a discussion on the potential for mediating or moderating variables that could provide deeper insights into the relationships being examined.

5. The results section presents significant differences across groups; however, the discussion lacks a critical interpretation of these findings. For instance, the authors assert that increased loneliness correlates with higher rejection sensitivity and avoidance behaviors, yet they do not address the implications of these relationships adequately. How might these cognitive and behavioral tendencies manifest in real-life social contexts? A more profound exploration of the practical implications of the findings would enhance the relevance of the research.

6. PLOS authors have the option to publish the peer review history of their article (what does this mean? ). If published, this will include your full peer review and any attached files.

**Do you want your identity to be public for this peer review?** For information about this choice, including consent withdrawal, please see our Privacy Policy .

Reviewer #1: No

Reviewer #2: No

Reviewer #3: No

Reviewer #4: No

Reviewer #5: **Yes: ** Chia-Ling Hsu

---

## [Author Response · Author response to Decision Letter 1]

7 Feb 2025

Point-by-point response

Comment from the authors:

We thank all reviewers very much for their time and their valuable and thoughtful comments on our manuscript. All revisions are highlighted in the manuscript. We wholeheartedly believe that the suggestions and our respective changes improved the manuscript significantly.

After implementing the comments we streamlined several parts of the manuscript and rearranged the results section to achieve a more logical structure. Further, we adjusted Table 4 by replacing the sum scores with the means and standard deviations of the lonely and non-lonely groups in terms of cognitive and behavioral aspects, since we believe it would be more intuitive information for the reader.

Reviewer #1:

Abstract:

Comment 1:

• This is just my personal opinion, but “ancova” as a keyword seems strange to me. It's not a specific modeling approach that people might be looking for, nor is it a content-related keyword.

• Response: We appreciate this suggestion and agree that "ancova" may not be an intuitive keyword for this manuscript. We have replaced it with “social cognition” - a more content-related term.

Introduction:

Comment 2:

• In the research I was involved in, we looked at the fact that loneliness should be differentiated at the state and trait level. This aspect could fit well in the introduction (only if the authors consider it relevant).

Gründahl, M., Weiß, M., Maier, L., Hewig, J., Deckert, J., & Hein, G. (2022). Construction and validation of a scale to measure loneliness and isolation during social distancing and its effect on mental health. Frontiers Psychiatry, 13, 798596.

• Response: Thank you for the suggestion. We appreciated looking into this research and acknowledge the importance of differentiating between state and trait loneliness. However, we refrained from introducing more terms regarding different aspects of loneliness but still incorporated this valuable paper in this section in the introduction:

“However, some scholars define maladaptive forms of loneliness based on frequency [e.g., 40], others based on distress associated with loneliness [e.g., 37], and others based on chronicity [e.g., 41]. To counter this heterogeneity, recent research has increasingly emphasized the importance of assessing loneliness as a multidimensional construct, incorporating different aspects [e.g., 31, 42]. These efforts align with the broader need to refine loneliness measurement and account for its diverse manifestations.”

Comment 3:

• Overall, I really liked the introduction very much. If the authors could think about how to tighten it up a bit, I think a lot of people might enjoy reading it. The rationale for the approach is well explained. It's a bit sad that the authors did not pre-register their study.

• Response: We appreciate the positive feedback on the introduction and have revised it to enhance conciseness while maintaining clarity (see highlighted sections in the manuscript). Regarding pre-registration, we agree and acknowledge this limitation and now mention it explicitly in the limitations section:

“Finally, this study it should be mentioned that this study was not pre-registered, which limits transparency in hypothesis testing.”

Methods:

Comment 4:

• Please provide further information on the survey: How many participants dropped out of the study? Was there a data-related quality/exclusion criterion (e.g., straightlining)? Were the questionnaires presented in random order or always in the same order (which order?)?

• Response: We appreciate the suggestion to provide additional details about the survey procedure and data quality control. We have now included information regarding participant dropout rates, exclusion criteria, and questionnaire presentation order in the Methods section:

“A total of 1,389 individuals initially accessed the online survey. Of these, 553 participants (39.8%) did not finish the survey and were excluded from the final sample. To ensure data integrity, bogus items were implemented in the survey, where another 23 participants were excluded. Two participants who reported being under the age of 18 were also excluded. Finally, we used complete-case analysis, leading to the exclusion of an additional 21 participants. The final sample consisted of 790 German-speaking adult participants.”

“All questionnaires were presented in a fixed order to maintain consistency. The order was as follows: (1) demographic information, (2) loneliness measures, (3) cognitive and behavioral variables, and (4) psychopathological.”

Results:

Comment 5:

• As the authors state in their introduction, loneliness can also maintain or further reinforce maladaptive cognitive and behavioral patterns. Since this study is cross-sectional, the direction is not clear. I wondered if adding three logistic regressions (one for each division method) using all the outcomes from the ancovas as predictors could also account for this aspect? This is by no means mandatory, it was just an idea that could also be argued against.

• Response: Thank you for the interesting suggestion to conduct additional logistic regressions to explore whether the cognitive and behavioral variables predict loneliness classifications. While we acknowledge the importance of examining potential directional influences, we decided not to include these additional analyses for several reasons:

o Our primary goal is to test the cognitive model of loneliness, not necessarily to determine the predictive strength of each component for loneliness classification. While predictive models can be valuable, our study is more concerned with exploring group differences based on loneliness dimensions, aligning with prior literature that takes a categorical approach.

o Since our study is cross-sectional, even logistic regressions would not establish causal directionality. While they could identify associations between cognitive/behavioral patterns and loneliness classifications, they would not clarify whether these variables drive loneliness or result from it. Without a longitudinal or experimental design, directionality remains speculative.

o Adding logistic regressions would increase statistical complexity without necessarily improving practical interpretability. Given the already comprehensive results presented, additional models could make the paper more cumbersome to read

Thus, while we recognize the merit of the reviewer’s suggestion, we opted to maintain our current analytic approach to keep the focus on group comparisons rather than predictive modeling. We acknowledge the study’s cross-sectional limitations and rewrote the first limitations section to acknowledge the suggestion and note that longitudinal studies would be better suited to address causality and directionality concerns.

“First, while our findings indicate significant differences in cognitive and behavioral processes across different loneliness classifications, the cross-sectional design does not allow for conclusions about causality. Future research should employ longitudinal designs to disentangle whether cognitive biases and behavioral tendencies contribute to loneliness persistence or emerge as a consequence of prolonged loneliness.”

Limitations:

Comment 6:

• I don't want to start an unrelated Covid debate here. But the question about the last 24 months in 2021/2022 could have led to a distorted self-assessment of chronic loneliness, as many people may have thought about lockdowns, etc. This could also (partly) explain the relatively high prevalence of chronic loneliness.

• Response: We appreciate the concern regarding the potential influence of COVID-19 on self-reports of chronic loneliness. The 24-month timeframe (2021/2022) coincided with the pandemic, raising the possibility that some individuals might have provided distorted perceptions of the duration of loneliness due to social distancing.

However, evidence suggests that the pandemic did not uniformly increase loneliness across all populations. For example, a meta-analysis by Ernst et al. (2022) found small but significant increases in loneliness prevalence during the pandemic. However, these increases varied across studies and populations, with some reporting no changes or even decreases in loneliness. This might suggest that while COVID-19 may have amplified loneliness for some, it is unlikely to be the sole explanatory factor for chronic loneliness prevalence.

Furthermore, chronic loneliness is not solely dependent on social isolation but is also shaped by stable individual factors, cognitive patterns, and perceived social disconnection. To acknowledge this potential bias, we have added a discussion to the Limitations section:

“Ninth, this study's assessment of chronic loneliness covered the past 24 months, a period that overlapped with the COVID-19 pandemic. This raises the possibility that some participants may have perceived the duration of their loneliness as chronic due to social distancing and lockdowns, even if it could have been primarily situational. However, research suggests that while overall loneliness increased during the pandemic, the effects were small [6]. Given that loneliness is influenced by stable cognitive and emotional factors rather than just external circumstances, the extent of this bias remains unclear. Future research should compare post-pandemic data to determine whether chronic loneliness rates have returned to pre-pandemic levels.”

Reviewer #2:

Comment 1:

• Lines 30-32: The statement on the division of the sample into "lonely and not lonely" individuals could be clarified. I suggest adding a more detailed description of the methodology used to distinguish between the different groups, specifying the criteria for each measure more clearly.

• Response: We are thankful for this suggestion. Given the space limitations of the abstract, we aim to provide a concise summary rather than a detailed methodological explanation. However, we have slightly clarified the wording to ensure the classification criteria are easily understood. A full description of the methodology is available in the Methods section.

“We classified 790 German-speaking adults (MAge = 31.86 (12.48), 81% female) as lonely or not lonely based on three dimensions - loneliness frequency, distress, and chronicity - and tested for group differences regarding cognitive and behavioral aspects, as proposed by the cognitive model of loneliness, while controlling for depressive and social anxiety symptoms.”

Comment 2:

• Lines 49-51: The control of depressive and social anxiety symptoms is mentioned. It would be helpful to expand this section by providing more details on the methods and tools used to control for these factors.

• Response: Thank you for your comment. We agree that providing more details on the methods used to control for depressive and social anxiety symptoms enhances clarity. To address this, we have expanded the relevant section to specify that depressive symptoms were assessed using the Patient Health Questionnaire-9 (PHQ-9), and social anxiety symptoms were measured using the Social Interaction Anxiety Scale-6 (SIAS-6). Both measures were included as covariates in our analyses to account for their potential influence on cognitive and behavioral differences in loneliness. This information has now been incorporated into the manuscript to ensure transparency regarding our methodological approach.

“Sixth, we performed three (for each classification method) two-sided ANCOVA to test group differences regarding interpretation bias, rejection sensitivity, social avoidance behavior, distress disclosure, self-esteem, and avoidance goal intensity while controlling for depressive (PHQ-9) and social anxiety symptoms (SIAS-6) by including them as continuous covariates. This approach allowed us to isolate the specific effects of loneliness facets on cognitive and behavioral variables and ensured that observed differences are not solely driven by underlying psychopathology.”

Comment 3:

• Lines 121-125: The definition of "chronic loneliness" could benefit from further elaboration, explaining the choice of the 24-month period as a threshold for chronicity, perhaps integrating additional studies that support this choice.

• Response: Thank you for your suggestion. We have clarified our operationalization of chronic loneliness in the method section. We based our definition on Young (1982), who distinguished chronic loneliness from transient loneliness by defining it as a state in which an individual has experienced loneliness for two or more years. However, while we followed Young’s definition, previous studies have also used the two-year threshold to define chronic loneliness, either through self-reported duration in cross-sectional studies (George, 1984; Gerson & Perlman, 1979; Hojat, 1983) or primarily in longitudinal designs where loneliness had to be present at two consecutive time points to be classified as chronic (e.g., Lim et al., 2023; Martin-María et al., 2021; Theeke, 2010). Due to the cross-sectional design, we used the former method and adapted it by incorporating the direct single-item with the threshold of feeling lonely at least “quiet often”. We contextualized this more in the introduction and the method sections:

Introduction:

“In the context of specifically investigating chronic loneliness, the definition of Young [43] has been used as a reference, which characterizes chronic loneliness as loneliness persisting for two years and beyond. Older cross-sectional studies relied on the self-reported duration of loneliness to operationalize chronic loneliness [44-46]. As previously mentioned, Qualter et al. [31] pointed out that most loneliness scales, which are frequently used in recent studies, typically ask participants about the frequency of their loneliness, using Likert-type scales from 'never' to 'always.’ These scales are then also applied in longitudinal studies to classify participants into 'not lonely,' 'temporary lonely,’ and 'chronically lonely' groups, where the occurrence of a certain level of loneliness at all time points determines the affiliation to one of the categories [e.g., 47, 48-52]. These study designs seem to integrate both duration and frequency aspects. However, one potential issue emerging from this is that it is unclear how well chronic loneliness is encompassed in these studies since changes in loneliness between the time points were not captured.”

Methods:

“We then created a dichotomous variable (1 = lonely and 0 = non-lonely) with a cut-off value of 24 months in accordance with the definition by Young [43], which has been used in previous cross-sectional [44-46] and longitudinal studies [e.g., 47, 48-52].”

Comment 4:

• Lines 206-210: The results regarding the agreement between the three loneliness measures (frequency, distress, chronicity) could be supported by explanatory graphs that visually illustrate the percentage of agreement and disagreement between the different measures.

• Response: Thank you for your suggestion. We have now added a bar chart that visually illustrates the percentage agreement between the different loneliness measures (frequency, distress, and chronicity). We believe that this addition makes the results more accessible and enhances the interpretation of our findings.

Comment 5:

• Lines 255-265: The interpretation of the correlation between loneliness and social avoidance behavior could be enriched by discussing possible psychological mechanisms that might explain this link.

• Response: Thank you for your suggestion. We have expanded the discussion of the link between loneliness and social avoidance behavior by integrating potential psychological mechanisms that may explain this relationship. Specifically, we now emphasize how interpretation bias and rejection sensitivity may contribute to social avoidance, which in turn limits opportunities for social connection and exacerbates loneliness.

“Lonely individuals tend to anticipate rejection, leading them to be more vigilant for social threats [79, 80], which is theorized to promote social withdrawal [10, 12]. Accordingly, our results show that loneliness comes with increased social avoidance behavior. This finding is in line with studies showing that lo

---

## [Decision Letter · Decision Letter 1]

14 Mar 2025

Revisiting the Cognitive and Behavioral Aspects of Loneliness: Insights from Different Measurement Approaches

PONE-D-24-42587R1

Dear Dr. Skoko,

We’re pleased to inform you that your manuscript has been judged scientifically suitable for publication and will be formally accepted for publication once it meets all outstanding technical requirements.

Kind regards,

Bao-Liang Zhong

Academic Editor

PLOS ONE

Additional Editor Comments (optional):

Reviewers' comments:

Reviewer's Responses to Questions

**Comments to the Author**

1. If the authors have adequately addressed your comments raised in a previous round of review and you feel that this manuscript is now acceptable for publication, you may indicate that here to bypass the “Comments to the Author” section, enter your conflict of interest statement in the “Confidential to Editor” section, and submit your "Accept" recommendation.

Reviewer #1: All comments have been addressed

Reviewer #2: All comments have been addressed

Reviewer #5: All comments have been addressed

2. Is the manuscript technically sound, and do the data support the conclusions?

Reviewer #1: Yes

Reviewer #2: Yes

Reviewer #5: Yes

3. Has the statistical analysis been performed appropriately and rigorously?

Reviewer #1: Yes

Reviewer #2: Yes

Reviewer #5: Yes

4. Have the authors made all data underlying the findings in their manuscript fully available?

Reviewer #1: Yes

Reviewer #2: Yes

Reviewer #5: Yes

5. Is the manuscript presented in an intelligible fashion and written in standard English?

Reviewer #1: Yes

Reviewer #2: Yes

Reviewer #5: Yes

6. Review Comments to the Author

Reviewer #1: The authors have responded to all comments thoroughly. Congratulations on this nice contribution to the literature.

Reviewer #2: We are therefore pleased to inform you that the manuscript, in its final version, is now ready to be accepted and published. We believe that this article represents a significant contribution to the field and will be of great interest to the readers.

Thank you for your commitment and collaboration during this revision process. Should there be any further steps required before official publication, we remain at your disposal for any final clarifications or adjustments.

Reviewer #5: Thank you for thoroughly addressing my previous questions, I generally agree with your responses. I have no more question.

7. PLOS authors have the option to publish the peer review history of their article (what does this mean? ). If published, this will include your full peer review and any attached files.

**Do you want your identity to be public for this peer review?** For information about this choice, including consent withdrawal, please see our Privacy Policy .

Reviewer #1: No

Reviewer #2: **Yes: ** Pierluigi Diotaiuti

Reviewer #5: No

---

## [Editor Report · Acceptance letter]

PONE-D-24-42587R1

PLOS ONE

Dear Dr. Skoko,

I'm pleased to inform you that your manuscript has been deemed suitable for publication in PLOS ONE. Congratulations! Your manuscript is now being handed over to our production team.

Kind regards,

on behalf of

Dr. Bao-Liang Zhong

Academic Editor

PLOS ONE